# Determinants of Changes in the Diet Quality of Japanese Adults during the Coronavirus Disease 2019 Pandemic

**DOI:** 10.3390/nu15010131

**Published:** 2022-12-27

**Authors:** Fumi Hayashi, Yukari Takemi

**Affiliations:** Faculty of Nutrition, Kagawa Nutrition University, 3-9-21 Chiyoda, Saitama 3500288, Japan

**Keywords:** COVID-19, diet quality, dietary consciousness, adults, Japan

## Abstract

This study evaluated changes in diet quality during the coronavirus disease 2019 (COVID-19) pandemic and its association with variations in attitudes or behavior, as well as health status changes. Participants were Japanese adults aged 20–69 years who resided in 13 prefectures where specific cautions were announced to prevent the spread of the virus. An online survey was conducted in September 2021, and participants were those who shopped for food or prepared meals more than twice a week during the survey. Overall, 2101 participants were analyzed. An improved or worsened diet quality was determined based on changes in food consumption patterns, and participants were categorized into three groups (improved diet quality (IDQ), worsened diet quality (WDQ), and others). The IDQ group participants (10.2%) improved their dietary consciousness during COVID-19, cooked almost everything from ingredients, and increased their balanced meal eating frequency. However, the WDQ participants (11.1%) had worsened dietary consciousness and increased consumption of takeaway meals and alcohol but decreased balanced meal consumption. Cooking frequency changes were not independent determinants of variations in diet quality. Our results show that the diet quality changes during COVID-19 were possibly caused by changes in dietary consciousness or different levels of meal preparation practices.

## 1. Introduction

The coronavirus disease 2019 (COVID-19) pandemic has affected food-related behavior in people in different ways. Although numerous studies reported that most people continued to engage in similar food-related behavior despite the COVID-19 pandemic [1,2,3], some people changed their food-related behavior in either a more favorable or unfavorable way. Several studies have reported positive changes in dietary habits, including an increased consumption of fresh produce (e.g., fruits and vegetables) and home cooking [4,5]. However, another systematic review of longitudinal studies that evaluated eating behavior changes during the COVID-19 pandemic reported that the outcomes were more about negative changes (e.g., consumption of increased amount of ultraprocessed foods, including snacks or sweets, increased amount of food eaten, decreased fruit and vegetables, and lower adherence to a healthy diet) than positive changes (e.g., adherence to a healthy diet) [6].

Having more time to prepare meals during the lockdown was perceived as the most important reason for a more balanced diet for the general French population [1]. Spending more time at home and remotely working from home leads to spending more time cooking or preparing family meals [7,8,9]. A study among Croatian adults showed that increased cooking frequency during confinement was associated with increased vegetable, legume, fish, and seafood consumption [7]. However, preparing dinners by buying ready-made meals has also been observed in several countries instead of consuming takeaway foods [9]. Because of these changes, weight gain has also been reported as a common negative health impact of home confinement in many studies [4,5].

Although Japan did not have as severe a lockdown as other countries, favorable and unfavorable eating behavior changes were reported as in other developed countries. Individuals whose dietary habits improved were healthier than those whose diets did not change during the COVID-19 pandemic [10]. Increased time spent at home also resulted in changes in cooking and dietary habits, as shown earlier [7,8,9]. Working from home during the government’s declaration period during the COVID-19 pandemic was significantly associated with increased consumption frequency of fruits, vegetables, and dairy products among Japanese adults [11]. The increased time spent at home due to COVID-19 led to opportunities to change cooking habits and dietary behaviors [12].

Domestic food preparations had been steadily declining before the pandemic. According to a nationwide survey on Japanese food culture [13], the percentage of adults who cook “almost every day” decreased by 0.8% (42.7–41.9%) from 2015 to 2019. Possible reasons for such a decline in home cooking in Japanese households are limited time [14,15], time consumption [15], and lower cooking skills [16,17,18]. In contrast, possible factors contributing to the high cooking frequency were confidence in cooking skills and positive perceptions of cooking (i.e., cooking is fun and not troublesome) [14,18,19]. Therefore, it is possible that the increased time at home enabled people to overcome some potential barriers to cooking since the additional time at home could be dedicated to enjoying cooking. However, the determinants associated with changes in the diet quality owing to the COVID-19 pandemic are yet to be determined.

Changes in food intake during the lockdown period due to the COVID-19 pandemic have also been reported in several countries [6,10,20,21]; however, most previous studies have examined changes by the food group and related factors. Only a few studies have evaluated changes in diet quality on an individual basis [22,23]. In Tribst et al.’s study [22], increased time spent on eating activities was positively associated with improved diet quality (IDQ) but negatively related to worsening diet quality. Bühlmeier et al. [23] found that individuals with health-oriented patterns increased consumption of healthy foods, time spent on cooking and eating, and abstinence from sweets, snacks, fast food, convenience products, and alcoholic beverages. Although these previous studies examined changes in the amount of time spent preparing meals, the associations between how meals were prepared, how the level of health literacy was affected, and how dietary attitudes or behavior changed are still yet to be adequately evaluated. Therefore, this study aimed to identify attitudes and behavior associated with changes in the diet quality during the COVID-19 pandemic.

## 2. Materials and Methods

### 2.1. Survey Methodology and Participants

A web-based survey was conducted from 1 September to 6 September 2021, to collect data on the Japanese population regarding changes in dietary attitudes and habits related to the COVID-19 outbreak. The participants were adults aged 20 to 69 years who resided in 13 prefectures (Tokyo, Kanagawa, Saitama, Chiba, Osaka, Hyogo, Fukuoka, Hokkaido, Ibaraki, Ishikawa, Gifu, Aichi, and Kyoto); these were designated as “special alert prefectures”, where it was necessary to promote efforts to prevent the spread of infections under the Japanese government from April to May 2020. They were recruited through a consumer panel managed by Intage Inc. (Tokyo, Japan). Since age and gender ratio-appropriate participants, based on the corresponding population ratio of each prefecture per the national census data by region, needed to be recruited, we decided to recruit them through Intage Inc., which has the largest number of consumer panels in Japan (as of 2020, holding approximately 4.5 million panelists). By using Intage Inc., we can avoid an exclusion bias and ensure that we have the required number of survey participants from a single provider, thus avoiding obtaining multiple responses by the same respondent. In the 2021 survey, we targeted 2000 participants (1000 men and women, and 400 each for the 10-year-old age groups of both sexes), which was the same as in the 2020 survey. In addition, a screening test was conducted on approximately 10,000 people to select participants who were involved in cooking or shopping for at least two days a week during the COVID-19 pandemic. The 2021 survey was conducted as a follow-up study in 2020. In the 2021 survey, a screening test was again administered to the panel, which included respondents of the 2020 survey and new respondents, and recruiting priority for the subsequent survey was given to those who had participated in the 2020 survey. Of the 3294 participants invited to participate in this survey, 2342 completed the survey. Overall, 2267 participants were considered eligible after excluding ineligible participants. To detect inconsistent responses, we inserted an inverted term in the questionnaire. If the participants had consistent answers throughout the questionnaire, we considered those responses as fraud and excluded from the analysis. After excluding those who indicated that they did not prepare their meals, 2101 respondents were included in the analysis.

### 2.2. Ethics Approval

Consent for study participation and data usage was obtained from Intage Inc. (Tokyo, Japan). Participants’ personal information, including their names, was anonymized to maintain and protect confidentiality. Therefore, the final database excluded personal data. The Ethics Committee of Kagawa Nutrition University approved this study protocol (Saitama, Japan; approval date: 17 March 2021).

### 2.3. Measures

The questionnaires included changes in the consumption of food groups during the COVID-19 outbreak, sociodemographic variables, subjective health status, physical variables, health literacy, and changes in dietary and other lifestyle behavior during the COVID-19 outbreak. The pre-COVID-19 period was defined as before February 2020. This is because the first COVID-19 patient was reported in Japan in February 2020, and hence, we defined the earlier period as before COVID-19.

#### 2.3.1. Changes in the Consumption of Food Groups

We measured changes in the consumption of 12 food groups (whole grains, fish and shellfish, lean meats, eggs, milk and dairy products, soy and soy products, green and yellow vegetables, other vegetables, seaweeds, mushrooms, potatoes, and fruits) that were recommended for daily consumption, and six food groups (processed meats or fish products, snacks and desserts, alcoholic beverages, sweetened beverages, frozen meals, and instant products) that were not recommended for daily consumption; this was according to the dietary guidelines for Japanese people and the concepts of the Japanese diet, which were recommended in the diet therapy guidelines for preventing atherosclerotic cardiovascular disease [24,25,26]. The response options were “increased”, “decreased”, and “no change”.

#### 2.3.2. Sociodemographic Variables

Sociodemographic variables included demographic questions (age, sex, marital status, household status, and living with children) and socioeconomic status (employment status, job status, highest educational qualification, annual household income, changes in household income compared to before COVID-19, household economic status, household economic status before COVID-19, and changes in household economic status compared to before COVID-19).

#### 2.3.3. Subjective Health Status and Physical Variables

Subjective health status during the survey was assessed using a single question, and the answers were excellent, good, poor, and very poor. Mental distress was assessed using the Kessler Psychological Distress Scale (K6), which is a 6-item inventory rated on a 5-point Likert-type scale [27]. The K6 cut point of 13 was developed to operationalize the definition of severe mental illness, followed by moderate (5 ≤ K6 < 13) and none or low mental distress (K6 < 5) [28]. Body mass index (BMI) categories were determined from calculated BMI based on self-reported height and weight, and cutoffs were based on the guidelines of the Japan Society for the Study of Obesity [29]. The cutoffs were as follows: underweight (<18.5 kg/m^2^), normal (18.5 ≤ BMI < 25.0 kg/m^2^), and overweight/obese (≥25.0 kg/m^2^). We also assessed changes in body weight and subjective health status during the COVID-19 pandemic.

#### 2.3.4. Health Literacy

Health literacy was measured using the Communicative and Critical Health Literacy scale developed by Ishikawa et al. [30]. This scale comprises five items addressing whether the respondent can (1) collect health information from various sources, (2) extract relevant information, (3) understand and communicate the information obtained, (4) consider the credibility of the information, and (5) make decisions based on the information in the context of health issues. Each item was scored on a 5-point scale ranging from strongly disagree (1 point) to strongly agree (5 points). The scores for the items in each scale were summed and divided by the number of items in that scale to yield a scaled score (theoretical range = 1–5). Cronbach’s alpha for the scale was 0.875.

#### 2.3.5. Dietary Attitudes and Behaviors and Other Lifestyle Behaviors

Regarding dietary attitudes and behaviors, participants were asked about their level of cooking practices during the survey, changes in dietary attitudes, and changes in several dietary behaviors during the COVID-19 outbreak.

Cooking practices were assessed by four levels as follows: “cook almost everything from ingredients”, “use some commercial foods”, “use many commercial foods”, and “prepare meals using commercial foods for everything”. We defined “commercial foods” as ready-to-eat, frozen, instant, and prepackaged foods. For the analyses, we combined the last two answers into one group (“use a lot of commercial foods”) because of the small number of responses.

Changes in dietary attitudes were measured using the Dietary Consciousness Scale, which is a validated questionnaire that comprises 12 questions categorized into 2 subscales: (1) importance of diet (seven items) and (2) precedence of diet (five items) [2,31]. Participants responded to the question regarding changes in dietary consciousness due to COVID-19 as follows: no change, 0 points; improved, +1 points; and worsened, −1 point. After calculating the total score for each subscale, those who scored ≥+1, ≤−1, and 0 point were grouped in “improved”, “worsened”, and “no change”, respectively.

Changes in dietary behaviors were assessed in terms of cooking frequency, frequency of eating out, takeaway, meal delivery, and eating a well-balanced Japanese diet, which was defined as eating a combination of staple, main, and vegetable dishes at least twice daily. Participants responded as “increased”, “decreased”, or “no change”.

The participants were asked about changes in alcohol intake and physical activity during the COVID-19 outbreak. For alcohol intake, we asked “Has your current alcohol consumption changed compared to before COVID-19?” For physical activity, we asked “Has your current status of walking and similar physical activity or exercise in your daily life changed compared to before COVID-19?”

### 2.4. Scoring Changes in Diet Quality

Changes in diet quality were assessed according to the categorization proposed by Tribst et al. [22]. Tribst et al. compared the two groups, IDQ vs. the general population and worsened diet quality (WDQ) vs. the general population, as independent analyses. Here, we categorized the participants into the following three groups: IDQ, WDQ, and others. Using Tribst’s method [22], the general population group would include those whose dietary habits did not change and those whose habits changed in the opposite direction. Here, to understand the characteristics of those who changed for the better or worse compared to those who did not change, which is most of the population, we categorized the population into the following three groups: those who observed only improvement (IDQ), those who observed only worsening (WDQ), and those who did not fall into either category (others).

Responses to changes in each food group consumption were coded as +1 (increased), −1 (decreased), and 0 (no change), and the participants were classified based on the following criteria. Participants were considered to have IDQ if they satisfied the following conditions: scores of vegetable (green and yellow vegetables, or other vegetables) and fruit consumption were ≥+1 point; the total change in the score of the recommended 12 food groups was ≥+1; the total change in the score of non-recommended items was from 0 to −6 points. Participants were considered to have WDQ if they satisfied the following conditions: the total change in the score of six non-recommended items was ≥+1 point; the total change in the score of the recommended food groups was ≤0. The remaining participants were considered others.

### 2.5. Statistical Analysis

For descriptive purposes, we first assessed the distribution of the changes for each food group. Subsequently, we compared sociodemographic variables, physical and lifestyle variables, health literacy, and changes in dietary attitudes and behavior across groups based on the changes in diet quality. Additionally, the variables among the three groups were compared using the Chi-square test and residual analysis. Multinomial logistic regression analyses using stepwise methods were performed to analyze factors that influenced changes in diet quality to estimate the odds ratio (95% confidence interval (CI)) of the IDQ and WDQ groups compared to the other groups. These analyses were adjusted for age, sex, household income changes, and household economic status before COVID-19. All analyses were conducted using the international business machine (IBM) statistical package for the social sciences (SPSS), version 28.0 (IBM Japan Ltd., Tokyo, Japan). Statistical significance was set at *p* < 0.05.

## 3. Results

### 3.1. Distribution of Changes in Diet Quality

Figure 1 shows the participants’ distribution of the changes in food consumption. As illustrated, most participants remained the same as before COVID-19. A Cronbach’s alpha test was conducted to evaluate the internal consistency of the items regarding changes in the consumption of food groups. The results confirmed the internal validity (alpha = 0.722).

### 3.2. Comparisons of Sociodemographic Variables

As presented in Table 1, the percentage of participants in each group was as follows: IDQ (10.2%, n = 214), WDQ (11.1%, n = 233), and others (78.7%, n = 1654). Significant differences were observed in sex, changes in household income compared to before the COVID-19, household socioeconomic status before COVID-19, and changes in household socioeconomic status compared to before COVID-19. Those in the IDQ group were mostly female (65.4%), and the household economic status was affluent before COVID-19 (37.9%). More people had a worsened household economic status compared to before COVID-19 had WDQ, although some also had an improved household income. The percentages of those whose household income decreased compared to that before COVID-19 were high in both the IDQ (33.2%) and WDQ (36.9%) groups.

### 3.3. Comparisons of Subjective Health Status and Physical Variables

Regarding subjective health status and physical variables (Table 2), significant differences were observed in the current subjective health status, mental distress, and changes in body weight and subjective health status. Participants categorized as IDQ had good current subjective health perception and improved their perception compared to before COVID-19. In addition, their body weights decreased. However, those in the WDQ group had poor subjective health status, moderate mental distress, worsened subjective health status, and increased body weight.

### 3.4. Comparisons of Health Literacy as Well as Dietary and Lifestyle Variables

Table 3 shows significant differences in all variables among the three groups. Those categorized as IDQ were high in health literacy, cooking almost everything from ingredients, improved consciousness on the importance and precedence of diet, and increased cooking and frequency of eating a balanced meal. In addition, participants in the IDQ group had a decreased frequency of eating out and alcohol consumption. Conversely, the opposite change was observed in those in the WDQ category. However, some variables (e.g., frequencies of takeaway and meal delivery) showed an increase as well as a decrease in such behavior in some people.

### 3.5. Factors Associated with the Pattern of Changes in Diet Quality

Table 4 shows the results of the multinomial logistic regression analysis of the factors associated with the pattern of changes in diet quality. Participants who cooked almost everything from ingredients, had improved consciousness on the importance and precedence of diet, and increased frequency of eating a balanced meal were more likely to enhance their diet quality than those in the other groups. However, participants with worsened consciousness on the importance and precedence of diet, decreased frequency of eating a balanced meal, and increased alcohol intake were more likely to have WDQ. The frequency of eating out decreased in both the IDQ and WDQ group compared to the participants in the other groups. Health literacy showed no significant association; however, participants with WDQ tended to have lower health literacy.

### 3.6. Participants’ Health Status with the Pattern of Changes in Diet Quality

Table 5 shows the results of the multinomial logistic regression analysis of the participants’ health status associated with the pattern of changes in diet quality. In the IDQ group, participants reported their subjective health status as both improved and worsened due to the COVID-19 pandemic; however, more of them had a good subjective health status during the survey. Conversely, in the WDQ group, more respondents had a worsened subjective health status.

## 4. Discussion

This study demonstrated changes in diet quality among Japanese adults during the COVID-19 pandemic and the factors associated with these changes. Approximately 20% of the participants changed their diet, and their diet either improved (10.2%) or worsened (11.1%) compared to before COVID-19. A previous study on Brazilians showed that approximately 20% of the population under study improved, and the other 10% worsened their diet quality [24]. Therefore, fewer participants in this study showed an improvement in dietary quality, although there were differences depending on the time of the survey. Participants with IDQ were more likely to cook almost everything from ingredients, and their dietary consciousness improved compared to before COVID-19. However, those with WDQ had a worsened dietary consciousness. A change in alcohol consumption was also an important factor associated with WDQ. The frequency of eating a well-balanced diet, which was defined as eating a combination of staple, main, and vegetable dishes, at least twice a day, was increased in those with IDQ but decreased in those with WDQ compared to participants in the others group. Previous studies have shown that people who increased their cooking frequency had better dietary intake than those who stayed unchanged or reduced the cooking frequency compared to before COVID-19 [2,7,32]. However, our results showed that diet quality was more associated with different levels of meal preparation practices rather than with simple increased cooking frequency.

Our study showed that participants categorized as IDQ also improved their dietary consciousness, both the importance and precedence, during the COVID-19 pandemic. In addition, those with worsened diet quality also had worsened dietary consciousness on both scales. In a previous study, those who changed their eating pattern to a health-oriented pattern during COVID-19 were paying more attention to the consumption of healthy foods than individuals who practiced emotion-driven eating behavior [23]. Another study showed that those with a high perception of the importance and precedence of diet were more likely to have been involved in home cooking during the COVID-19 pandemic [2]. In that study, participants whose cooking time and efforts increased during the COVID-19 pandemic had significantly higher scores for food groups recommended for daily consumption than participants with either decreased or unchanged cooking time and efforts.

Although the previous study did not assess diet in the context of its changes during the COVID-19 pandemic, this study further suggests that increased dietary consciousness is an essential determinant for improving diet quality. However, those with improved consciousness on the precedence of diet had 1.62 times more chance to worsen the diet quality than the group of others. The precedence of the diet subscale is not limited to healthy eating. However, even among those engaged in less healthy eating behavior, many indicated that they had become more concerned about the diet itself [23]. Therefore, it is possible that the participants prioritized diet during the COVID-19 pandemic; however, those with WDQ did not consider their nutrient content or health issues. 

Although not statistically significant, those with higher health literacy scores tended to have a lower chance of worsening dietary quality. Health and food literacy are associated with healthy eating habits [33,34,35,36]. A significant relationship was found between healthy eating literacy scores and a well-balanced diet among Japanese adults. The higher the score, the more likely they were to eat a well-balanced diet frequently [33]. Another study reported that those with high health literacy tended to have a higher intake of vegetables and a lower sodium/potassium ratio than those with low health literacy [34]. Therefore, limited health literacy might worsen dietary quality. Nutrition and diet advice for the population during the COVID-19 outbreak has been released by many governments and professional organizations, among others. For example, the World Health Organization released nutrition advice urging people to eat fresh and unprocessed foods daily to be healthier with stronger immune systems and a lower risk of chronic illnesses and infectious diseases [37]. The Japanese government also released tips to the public regarding healthy eating habits and a new lifestyle [38]. In the government’s messages, there was an alarming message that the increased use of easily available processed foods and ready-made meals due to restrictions on going out would lead to excessive salt and fat intake. Health literacy to obtain such information and implement it is essential for acquiring desirable dietary habits. Therefore, nutrition education to enhance health literacy is crucial to achieving healthier eating.

Furthermore, the odds ratio for the IDQ of participants who increased the frequency of having a balanced meal was 2.68 higher than those with no change. In addition, an increase in balanced meals prevented diet quality from worsening; however, a decreased frequency of having a balanced meal increased the chance of worsening diet quality. Eating a well-balanced diet with a combination of staple, main, and vegetable dishes at least twice a day is one of the objectives of the national health promotion program in Japan [39]. Therefore, it was confirmed again that balancing the diet was the key to improving it. According to a previous study on pregnant Japanese women, the chance of eating a balanced meal more than twice a day was higher among those who valued cooking and were more confident in preparing nutritious meals [40]. Brasington et al. [41] found that adults who were less confident in cooking were more likely to use convenient cooking products, including meal/recipe bases (concentrates) or simmer sauces. Lam and [42] also reported that individuals with more confidence in their cooking skills used ultraprocessed foods less frequently. Furthermore, Metcalfe et al. [43] reported that improved cooking confidence was associated with improved vegetable and fruit intake, although the cooking frequency did not change. Lavelle et al. [44] highlighted that food skills (e.g., planning meals, knowing the food budget, and cooking meals with limited time) were more critical for diet quality than just cooking skills. Cooking is a complex behavior, and its relationship with meal quality can vary significantly depending on what and how it is prepared [45]. Therefore, in addition to simply encouraging more frequent cooking at home, improved cooking skills and confidence in cooking meals with fresh ingredients are required to achieve a more desirable diet.

Here, those who prepared meals with almost everything from ingredients were twice as likely to improve diet quality than those who cooked using a lot of commercial foods. However, the change in cooking frequency was not an independent determinant of the changes in diet quality. Previous studies have reported that an increased frequency of home cooking was associated with better diet quality [22,32,46]; however, those who used more convenient foods ate fewer vegetables [41]. Furthermore, Nakayasu et al. [32] highlighted the effect of infrequent cooking as a reason for the unfavorable change in eating behavior, even though cooking frequency increased during the COVID-19 pandemic. Specifically, an increased cooking frequency may not lead to a well-balanced diet if the actual cooking frequency is low. Regarding actual cooking frequency, more than half of the participants in the IDQ group cooked almost daily. In contrast, significantly fewer participants in the WDQ group cooked almost daily (data not shown). Thus, it was suggested that the change in cooking frequency and how they cooked meals was more important for better adherence to a well-balanced diet. Therefore, nutrition education should teach cooking skills and provide support to enhance comprehensive skills and develop confidence in preparing a desirable diet.

Here, an association was observed between the frequency of takeaway meals and changes in diet quality. The diet of participants who increased the frequency of takeaway meals worsened. According to a nationwide study among Japanese adults, those who consumed takeaway meals more frequently had lower vegetable intake and higher oil and fat intake [47]. In addition, a previous study reported that a high frequency of consuming takeaway meals was associated with nutrient inadequacy, including protein, dietary fiber, and potassium. A systematic review reported that a higher frequency of consuming meals prepared outside of home resulted in a lower intake of micronutrients, including vitamin C, calcium, and iron [48]. Moreover, those who consumed more homecooked meals had better eating habits; however, those who consumed more foods prepared outside of home, particularly fast food, were unhealthy [49]. Regarding the frequency of eating out, many respondents indicated that they ate out less regularly in both groups. The Family Income and Expenditure Survey of Japan reported a decrease in spending on eating out compared with that before the COVID-19 pandemic, regardless of the type of household [50]. However, the amount spent on prepared foods has been increasing since before the COVID-19 pandemic. In contrast, the amount spent on prepared foods has remained constant and is increasing even though the amount spent on eating out has decreased. Our results suggested that diet quality improved for those who ate out less and cooked more from ingredients at home, whereas diet quality worsened for those who replaced food from eating out with prepared meals.

Changes in alcohol consumption and subjective health status were associated with changes in diet quality. The IDQ group had decreased alcohol consumption, and most of them had a good subjective sense of health, although both changes were observed. However, participants in the WDQ group had increased alcohol consumption and worsened their subjective sense of health. A previous study showed that more health-conscious participants during the COVID-19 pandemic consumed fewer alcoholic beverages. However, those who paid less attention to healthy eating consumed more alcoholic beverages [23]. According to a longitudinal study on working-age Finnish participants, a positive change in health behavior, including dietary and alcohol intake, predicted better subjective well-being; however, a negative change predicted poorer subjective well-being [51]. Older people with higher subjective well-being also demonstrated better diet quality and eating behavior [52]. Therefore, our results support the findings of the previous studies. Furthermore, increased alcohol intake in the context of poorer diet quality could increase the risk of hypertension [53]. Therefore, changes in dietary and health behavior should have been carefully monitored during the COVID-19 pandemic and a support system established for those at high risk.

This study had some limitations. First, this was a cross-sectional study with no random sampling, and the participants were recruited from the panel of an internet research company. A web-based questionnaire with volunteer panels could lead to selection and response biases [54,55]. According to the 2020 Ministry of Internal Affairs and Communications statistics, approximately 20% of the Japanese population still lacks Internet access [56]. In addition, the Internet panel tended to include more women, older aged individuals, and those residing in large cities [55]. However, based on recent population census data, we invited participants by considering the age group and sex ratio of each prefecture. Second, dietary habits and intake changes were self-reported, which subjected them to error and bias. In addition, we classified frozen foods into the unhealthy food groups. However, frozen foods also include unprocessed foods such as vegetables and seafood. According to statistics [57], since frozen ready foods or meals account for more than 95% of the total volume of frozen foods supplied to Japanese households, we classified frozen foods into the unhealthy food group. Further research is needed to assess actual food intake to enhance the validity of food and nutrient intake or its changes. Third, differences in the timing of the surveys may have led to different results from previous studies [22]. This study was conducted in September 2021, a time when the number of infected patients was relatively high in Japan. During the survey period, emergency declarations and measures to prevent the spread of the disease were issued in 21 prefectures throughout Japan, including prefectures subject to the first emergency declaration [58]. Although there may have been less urgency than at the beginning of the outbreak in 2020, individuals’ dietary behavior might have been continuously influenced by the COVID-19 pandemic since many worked remotely, and supermarkets and restaurants had been under voluntary restraints on business and shortened operating hours at the government’s request. Although the status of remote work was not significantly different among groups, differences in the degree of restrictions by individuals may have affected the results and thus represent a limitation. Compared to a previous study [22], the proportions of those whose diets improved or worsened differed; however, it was confirmed that those whose diets improved had better food group intake than those whose diets worsened (Appendix A). Despite these limitations, this study reveals that more attention should have been paid to how meals are prepared, not just the frequency of home cooking, and improved dietary consciousness is an essential determinant for enhancing diet quality among the population during the COVID-19 pandemic.

## 5. Conclusions

Although previous studies have reported that increased frequency of home cooking was associated with a better diet owing to restricted outings during the COVID-19 pandemic, a change in cooking frequency was not an independent determinant of improvement in diet quality. Rather, this study demonstrated that the methods for preparing meals and improving dietary consciousness were important determinants of changes in diet quality. Our study suggests that increased dietary consciousness is associated with nutrition in cooking practices. In addition, confidence in preparing a meal from ingredients increases the frequency of well-balanced meals, increasing the quality of the diet. Furthermore, IDQ was associated with better subjective well-being. The study also suggests that health literacy prevents the worsening of dietary quality. Therefore, adherence to dietary recommendations requires increased home cooking practices, and how the meal is prepared is crucial.

## Figures and Tables

**Figure 1 nutrients-15-00131-f001:**
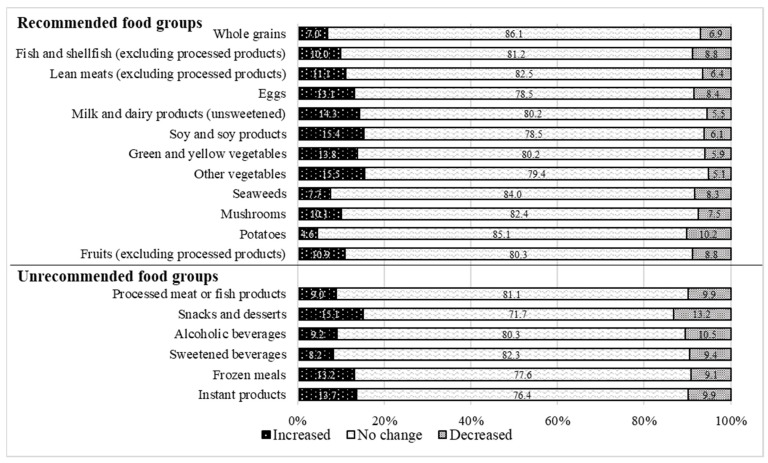
Percentage of participants reporting increased, no change, or decreased frequency of intake of 18 food groups (12 recommended and 6 unrecommended) compared to before COVID-19.

**Table 1 nutrients-15-00131-t001:** Characteristics of participants according to changes in diet quality.

		Total (*N* = 2101)	IDQ (*n* = 214)		WDQ (*n* = 233)		Others (*n* = 1654)		
		*N*	*n*	%	*n*	%	*n*	%	*p*
Sex	Male	971	74 *	34.6	97	41.6	800 **	48.4	<0.001
	Female	1130	140 **	65.4	136	58.4	854 *	51.6	
Age group, years	20–29	404	56	26.2	39	16.7	309	18.7	0.080
	30–39	412	47	22.0	47	20.2	318	19.2	
	40–49	427	34	15.9	47	20.2	346	20.9	
	50–59	423	33	15.4	56	24.0	334	20.2	
	60–69	435	44	20.6	44	18.9	347	21	
Marital status	Unmarried	767	72	33.6	81	34.8	614	37.1	0.352
	Married	1150	129	60.3	129	55.4	892	53.9	
	Divorced or widowed	184	13	6.1	23	9.9	148	8.9	
Household status	Living alone	507	44	20.6	53	22.7	410	24.8	0.346
	Two or more	1594	170	79.4	180	77.3	1244	75.2	
Living with children aged 15 years or younger	No	1659	170	79.4	175	75.1	1314	79.4	0.310
	Yes	442	44	20.6	58	24.9	340	20.6	
Employment status	Permanent employees	780	68	31.8	81	34.8	631	38.1	0.236
	Contract employees	114	11	5.1	8	3.4	95	5.7	
	Part-time workers	372	36	16.8	44	18.9	292	17.7	
	Self-employed	161	15	7.0	16	6.9	130	7.9	
	Students	53	10	4.7	3	1.3	40	2.4	
	Housewives	379	47	22.0	49	21.0	283	17.1	
	Unemployed	239	27	12.6	31	13.3	181	10.9	
	Others	3	0	0.0	1	0.4	2	0.1	
Job status	Fully remote working	134	19	14.6	11	7.3	104	9.0	0.059
	More remote working than working in the office	138	13	10.0	17	11.3	108	9.4	
	More working in the office than remote working	130	8	6.2	10	6.6	112	9.7	
	Fully working in the office	945	80	61.5	105	69.5	760	65.9	
	Currently not working	46	9	6.9	5	3.3	32	2.8	
	Don’t want to answer	41	1	0.8	3	2.0	37	3.2	
Highest educational qualification	Junior high school/high school	575	47	22.0	54	23.2	474	28.7	0.208
	Professional school/junior college	483	55	25.7	66	28.3	362	21.9	
	University	911	101	47.2	97	41.6	713	43.1	
	Graduate school	94	8	3.7	12	5.2	74	4.5	
	Don’t want to answer	38	3	1.4	4	1.7	31	1.9	
Annual household income, Yen	<2,000,000	338	28	13.1	39	16.7	271	16.4	0.465
	2,000,000–4,000,000	409	36	16.8	41	17.6	332	20.1	
	4,000,000–6,000,000	402	47	22.0	51	21.9	304	18.4	
	≥6,000,000	583	60	0.0	57	24.5	466	28.2	
	Don’t know/don’t want to answer	369	43	20.1	45	19.3	281	17.0	
Changes in household income compared to before COVID-19	Increased	72	12	5.6	7	3.0	53	3.2	<0.001
	Decreased	571	71 **	33.2	86 **	36.9	414 *	25	
	No change	1458	131 *	61.2	140 *	60.1	1187 **	71.8	
Household economic status	Not affluent	844	83	38.8	108	46.4	653	39.5	0.341
	Neither	679	68	31.8	69	29.6	542	32.8	
	Affluent	578	63	29.4	56	24.0	459	27.8	
Household economic status before COVID-19	Not affluent	614	54	25.2	76	32.6	484	29.3	0.027
	Neither	877	79	36.9	89	38.2	709 **	42.9	
	Affluent	610	81 **	37.9	68	29.2	461 *	27.9	
Changes in household economic status compared to before COVID-19	Worsened	467	58	27.1	70 **	30.0	339 *	20.5	<0.001
	Improved	208	21	9.8	34 **	14.6	153	9.3	
	No change	1426	135	63.1	129 *	55.4	1162 **	70.3	

IDQ—improved diet quality, WDQ—worsened diet quality; Chi-square test. ** Adjusted residual > 1.96, * Adjusted residual < −1.96.

**Table 2 nutrients-15-00131-t002:** Subjective health status and physical variables of participants according to changes in the diet quality.

		Total (*N* = 2101)	IDQ (*n* = 214)		WDQ (*n* = 233)		Others (*n* = 1654)		
		*N*	*n*	%	*n*	%	*n*	%	*p*
Current subjective health status	Excellent	174	25	11.7	13	5.6	136	8.2	<0.001
	Good	1405	161 **	75.2	140 *	60.1	1104	66.7	
	Poor	421	24 *	11.2	61 **	26.2	336	20.3	
	Very poor	101	4 *	1.9	19 **	8.2	78	4.7	
Body mass index category *	BMI < 18.5	322	33	15.4	37	15.9	252	15.2	0.895
	18.5 ≤ BMI < 25.0	1374	143	66.8	157	67.4	1074	64.9	
	BMI ≥ 25.0	345	32	15.0	35	15.0	278	16.8	
	Don’t want to answer	60	6	2.8	4	1.7	50	3.0	
Mental distress	Serious	217	15	7.0	28	12.0	174	10.5	0.010
	Moderate	808	90	42.1	108 **	46.4	610 *	36.9	
	None to low	1076	109	50.9	97 *	41.6	870 **	52.6	
Changes in body weight compared to before COVID-19	Increased	569	64	29.9	77 **	33.0	428 *	25.9	<0.001
	Decreased	310	46 **	21.5	43	18.5	221 *	13.4	
	No change	1109	99 *	46.3	105 *	45.1	905 **	54.7	
	Don’t know	113	5 *	2.3	8	3.4	100 **	6.0	
Changes in subjective health status	Improved	167	45 **	21.0	16	6.9	106 *	6.4	<0.001
	Worsened	339	34	15.9	69 **	29.6	236 *	14.3	
	No change	1595	135 *	63.1	148 *	63.5	1312 **	79.3	

Measured by Kessler Psychological Distress Scale (K6): serious (K6 ≥ 13), moderate (K6 5-12), none to low (K6 < 5). IDQ—improved diet quality, WDQ—worsened diet quality; Chi-square test. ** Adjusted residual > 1.96, * Adjusted residual < −1.96.

**Table 3 nutrients-15-00131-t003:** Health literacy and current level of cooking practices as well as changes in dietary and lifestyle variables among participants according to changes in the diet quality.

		Total (*N* = 2101)	IDQ (*n* = 214)		WDQ (*n* = 233)		Others (*n* = 1654)		
		*N*	*n*	%	*n*	%	*n*	%	*p*
**Health literacy and dietary attitudes**									
Health literacy	High	1162	147 **	68.7	117	50.2	898	54.3	<0.001
	Low	939	67 *	31.3	116	49.8	756	45.7	
Importance of diet	Improved	1001	177 **	82.7	117	50.2	707 *	42.7	<0.001
	Worsened	174	10 *	4.7	40 **	17.2	124 *	7.5	
	No change	926	27 *	12.6	76	32.6	823 **	49.8	
Precedence of diet	Improved	695	128 **	59.8	91 **	39.1	476 *	28.8	<0.001
	Worsened	308	38	17.8	57 **	24.5	213 *	12.9	
	No change	1098	48 *	22.4	85 *	36.5	965 **	58.3	
**Dietary Behaviors**									
Level of cooking practices	Cook almost everything from ingredients	725	95 **	44.4	61 *	26.2	569	34.4	<0.001
	Use some commercial foods	971	98	45.8	120	51.5	753	45.5	
	Use a lot of commercial foods	405	21 *	9.8	52	22.3	332	20.1	
Cooking frequency	Increased	778	146 **	68.2	90	38.6	542 *	32.8	<0.001
	Decreased	134	10	4.7	40 **	17.2	84 *	5.1	
	No change	1189	58 *	27.1	103 *	44.2	1028 **	62.2	
Frequency of eating out	Increased	91	10	4.7	11	4.7	70	4.2	<0.001
	Decreased	1087	161 **	75.2	169 **	72.5	757 *	45.8	
	No change	923	43 *	20.1	53 *	22.7	827 **	50.0	
Frequency of takeaway	Increased	479	71 **	33.2	85 **	36.5	323 *	19.5	<0.001
	Decreased	279	51 **	23.8	41 **	17.6	187 *	11.3	
	No change	1343	92 *	43.0	107 *	45.9	1144 **	69.2	
Frequency of meal delivery	Increased	307	51 **	23.8	49 **	21.0	207 *	12.5	<0.001
	Decreased	176	32 **	15.0	30 **	12.9	114 *	6.9	
	No change	1618	131 *	61.2	154 *	66.1	1333 **	80.6	
Frequency of eating a balanced meal	Increased	342	98 **	45.8	25 *	10.7	219 *	13.2	<0.001
	Decreased	182	15	7.0	49 **	21.0	118 *	7.1	
	No change	1577	101 *	47.2	159 *	68.2	1317 **	79.6	
**Other lifestyle behaviors**									
Alcohol intake	Increased	250	23	10.7	55 **	23.6	172	10.4	<0.001
	Decreased	331	63 **	29.4	31	13.3	237	14.3	
	No change	1520	128	59.8	147	63.1	1245 **	75.3	
Physical activity	Increased	218	37 **	17.3	24	10.3	157	9.5	<0.001
	Decreased	563	78 **	36.4	87	37.3	398	24.1	
	No change	1320	99	46.3	122	52.4	1099 **	66.4	

IDQ—improved diet quality, WDQ—worsened diet quality; Chi-square test. ** Adjusted residual > 1.96, * Adjusted residual < −1.96.

**Table 4 nutrients-15-00131-t004:** Multinomial logistic regression analyses of the factors associated with changes in the diet quality.

		IDQ (*n* = 214) vs. Others			WDQ (*n* = 233) vs. Others		
		OR	95% CI	*p*	OR	95% CI	*p*
**Health literacy and dietary attitudes**							
Health literacy	High	1.29	0.93–1.80	0.130	0.75	0.56–1.00	0.053
	Low	1.00			1.00		
Importance of diet	Improved	2.99	1.84–4.85	<0.001	1.10	0.75–3.05	0.628
	Worsened	1.34	0.59–3.02	0.483	1.82	1.08–3.05	0.024
	No change	1.00			1.00		
Precedence of diet	Improved	2.31	1.53–3.48	<0.001	1.62	1.11–2.38	0.013
	Worsened	1.56	0.93–2.60	0.092	2.43	1.57–3.78	<0.001
	No change	1.00			1.00		
**Dietary Behaviors**							
Level of cooking practices	Cook almost everything from ingredients	2.00	1.18–3.39	0.010	0.92	0.60–1.41	0.693
	Use some commercial foods	1.48	0.88–2.48	0.140	1.17	0.80–1.71	0.432
	Use a lot of commercial foods	1.00			1.00		
Cooking frequency	Increased	-			-		
	Decreased	-			-		
	No change	-			-		
Frequency of eating out	Increased	1.23	0.54–2.77	0.623	1.03	0.47–2.26	0.942
	Decreased	2.21	1.46–3.34	<0.001	2.67	1.83–3.89	<0.001
	No change	1.00			1.00		
Frequency of takeaway	Increased	1.20	0.81–1.78	0.373	1.71	1.19–2.46	0.003
	Decreased	1.35	0.87–2.10	0.186	1.18	0.75–1.85	0.473
	No change	1.00			1.00		
Frequency of meal delivery	Increased	-			-		
	Decreased	-			-		
	No change	-			-		
Frequency of eating a balanced meal	Increased	2.68	1.88–3.81	<0.001	0.53	0.33–0.87	0.011
	Decreased	1.18	0.64–2.19	0.600	2.21	1.44–3.39	<0.001
	No change	1.00			1.00		
**Other lifestyle behaviors**							
Alcohol intake	Increased	0.62	0.37–1.03	0.067	1.90	1.29–2.80	0.001
	Decreased	1.32	0.91–1.92	0.148	0.63	0.40–0.98	0.040
	No change	1.00			1.00		
Physical activity	Increased	-			-		
	Decreased	-			-		
	No change	-			-		

IDQ—improved diet quality, WDQ—worsened diet quality; independent variables are those listed in the table and sex, age groups, household income change, and household economic status before COVID-19. The independent variables were selected using a stepwise method.

**Table 5 nutrients-15-00131-t005:** Multinomial logistic regression analyses of participants’ health status associated with changes in diet quality.

		IDQ (*n* = 214) vs. Others			WDQ (*n* = 233) vs. Others		
		OR	95% CI	*p*	OR	95% CI	*p*
Current subjective health status	Excellent	3.24	1.05–9.96	0.040	0.53	0.24–1.17	0.118
	Good	2.90	1.03–8.19	0.045	0.67	0.39–1.18	0.164
	Poor	1.38	0.46–4.13	0.561	0.76	0.43–1.36	0.355
	Very poor	1.00			1.00		
Body mass index category	BMI < 18.5	-			-		
	18.5 ≤ BMI < 25.0	-			-		
	BMI ≥ 25.0	-			-		
	Don’t want to answer	-			-		
Mental distress	Serious	-			-		
	Moderate	-			-		
	None to low	-			-		
Changes in body weight compared to before the COVID-19	Increased	-			-		
	Decreased	-			-		
	Don’t know	-			-		
	No change	-			-		
Changes in subjective health status	Improved	3.74	2.49–5.59	<0.001	1.38	0.79–2.42	0.262
	Worsened	1.76	1.14–2.71	0.011	2.20	1.55–3.13	<0.001
	No change	1.00			1.00		

IDQ—improved diet quality, WDQ—worsened diet quality; independent variables are those listed in the table and sex, age groups, household income change, and household economic status before COVID-19. The independent variables were selected using a stepwise method.

## Data Availability

Data sharing is not applicable.

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
