# Peer review of "Determinants of Changes in the Diet Quality of Japanese Adults during the Coronavirus Disease 2019 Pandemic"

_nutrients, 2022, doi:10.3390/nu15010131_

Round 1

Reviewer 1 Report

The authors conducted a cross-sectional study to investigate attitudes and behaviors associated with changes in the quality of dietary patterns in a cohort of Japanese adults during the COVID-19 pandemic. This is an interesting study and the manuscript is well written. Please see my comments below and hopefully the authors can address my comments satisfactorily.

Major points

1. The authors should clearly explain how they determined/calculated the target sample size in the Methods section.

2. Lines 120-121, can the authors please comment on what they mean by “participants who did not correctly answer multiple questions”? The exclusion criteria should be explained in more detail in the Methods section.

3. Why did the authors decide to recruit participants through a consumer panel managed by Intage Inc? Can the authors explain what this consumer panel was? This will give the audience a better understanding of the recruitment process and any potential bias associated with this recruitment approach.

4. Lines 264-265, “However, frequencies of…increased and decreased” this sentence is confusing and needs to be rephrased to improve readability. In addition, results presented in Table 3 should be interpreted more clearly in section 3.4.

Minor points

5. The introduction is too long and can be shortened to make it more focused.

6. Line 145-146: results from the Cronbach’s alpha test should be moved to the Results section of the manuscript.

Reviewer 2 Report

I found referring to this as being about 'dietary patterns' confusing.  To me a dietary pattern is either a score derived from factor analysis/PCA ("a posteriori" patterns) or a score on a predefined index like DASH ("a priori").  While this sort of uses a 'compliance index' with Japanese dietary guidelines, there is also  singling out particular groups (fruit/vegetables). So this is really scores on 2 patterns (1 of healthy foods, one of unhealthy foods), plus a key indicator (fruit/veg). I realise this is largely inherited from the methodology of Tribst et al, but I think a little more explanation around this early on could go a long way. "Pattern" as it is used is quite vague, in particular I think simply referring to "diet quality" in the title would be better. 

a small note: line 183 refers to "frozen" foods; Tribst refers specifically to "frozen ready meals"--if that is the case here I would make it clear. Many frozen but otherwise unprocessed foods (eg frozen peas)  would probably be considered as "ingredients" rather than "commercial foods"

Reviewer 3 Report

1. Restrictions during COVID are considered as a reason for dietary changes. However, it does not describe the specific limitations of the individuals who participated in the survey. Different countries had different restrictions. What kind of restrictions were typical for 13 Japanese prefectures and for study participants: complete isolation, restriction of movement, remote work, restriction of access to shops, restaurants? Perhaps the presence of certain limitations was a criterion for inclusion in the study? It is necessary to describe in detail what limitations are implied in the study.

2. In which prefectures was the survey conducted?

3. It is necessary to briefly describe what is this “consumer panel managed by Intage Inc.”? Are there any features of these consumers?

4. It is necessary to describe very clearly and in detail the procedure for selecting participants in the study. I understand that in 2020 a screening test was conducted, and in 2021 2101 respondents were selected from these people? Or was the selection of study participants conducted in some other way?

5. For what period were changes in indicators in dynamics assessed (economic status, body mass index, subjective health status)? Did you have any specific dates in mind?

6. What method was used to assess physical activity?

7. In figure 1, the percentages should be indicated.

Round 2

Reviewer 1 Report

The authors have revised and improved their manuscript and adequately addressed my comments.